# The Feasibility of Using the “Artery Sign” for Pre-Procedural Planning in Navigational Bronchoscopy for Parenchymal Pulmonary Lesion Sampling

**DOI:** 10.3390/diagnostics12123059

**Published:** 2022-12-06

**Authors:** Elliot Ho, Roy Joseph Cho, Joseph C. Keenan, Septimiu Murgu

**Affiliations:** 1Division of Pulmonary & Critical Care Medicine, Interventional Pulmonology, Department of Medicine, Loma Linda University, 11234 Anderson St., Loma Linda, CA 92354, USA; 2Division of Pulmonary & Critical Care Medicine, Interventional Pulmonology, Department of Medicine, University of Minnesota, Minneapolis, MN 55455, USA; 3Division of Pulmonary & Critical Care Medicine, Interventional Pulmonology, Department of Medicine, University of Chicago, Chicago, IL 60637, USA

**Keywords:** robotic bronchoscopy, navigational bronchoscopy, bronchus sign, vessel sign, artery sign, pulmonary vessel, pulmonary nodules, lung cancer

## Abstract

Background: Electromagnetic navigation bronchoscopy (ENB) and robotic-assisted bronchoscopy (RAB) systems are used for pulmonary lesion sampling, and utilize a pre-procedural CT scan where an airway, or “bronchus sign”, is used to map a pathway to the target lesion. However, up to 40% of pre-procedural CT’s lack a “bronchus sign” partially due to surrounding emphysema or limitation in CT resolution. Recognizing that the branches of the pulmonary artery, lymphatics, and airways are often present together as the bronchovascular bundle, we postulate that a branch of the pulmonary artery (“artery sign”) could be used for pathway mapping during navigation bronchoscopy when a “bronchus sign” is absent. Herein we describe the navigation success and safety of using the “artery sign” to create a pathway for pulmonary lesion sampling. Methods: We reviewed data on consecutive cases in which the “artery sign” was used for pre-procedural planning for conventional ENB (superDimension™, Medtronic) and RAB (Monarch™, Johnson & Johnson). Patients who underwent these procedures from July 2020 until July 2021 at the University of Minnesota Medical Center and from June 2018 until December 2019 at the University of Chicago Medical Center were included in this analysis (IRB #19-0011 for the University of Chicago and IRB #00013135 for the University of Minnesota). The primary outcome was navigation success, defined as successfully maneuvering the bronchoscope to the target lesion based on feedback from the navigation system. Secondary outcomes included navigation success based on radial EBUS imaging, pneumothorax, and bleeding rates. Results: A total of 30 patients were enrolled in this analysis. The median diameter of the lesions was 17 mm. The median distance of the lesion from the pleura was 5 mm. Eleven lesions were solid, 15 were pure ground glass, and 4 were mixed. All cases were planned successfully using the “artery sign” on either the superDimension™ ENB (n = 15) or the Monarch™ RAB (n = 15). Navigation to the target was successful for 29 lesions (96.7%) based on feedback from the navigation system (virtual target). Radial EBUS image was acquired in 27 cases (90%) [eccentric view in 13 (43.33%) and concentric view in 14 patients (46.66%)], while in 3 cases (10%) no r-EBUS view was obtained. Pneumothorax occurred in one case (3%). Significant airway bleeding was reported in one case (3%). Conclusions: We describe the concept of using the “artery sign” as an alternative for planning EMN and RAB procedures when “bronchus sign” is absent. The navigation success based on virtual target or r-EBUS imaging is high and safety of sampling of such lesions compares favorably with prior reports. Prospective studies are needed to assess the impact of the “artery sign” on diagnostic yield.

## 1. Introduction

Diagnostic procedures for sampling parenchymal pulmonary lesions (PPL) have advanced remarkably over the last two decades as more lung nodules are being identified due to liberalized lung cancer screening guidelines, increasing prevalence of chronic lung disease, and improvements in advanced chest imaging [1]. Bronchoscopy is typically the preferred technique in the evaluation of PPL in patients with suspected lung cancer [2]. Advances in bronchoscopic methods for PPL sampling have been made in the field of electromagnetic navigation bronchoscopy (ENB) and robotic-assisted bronchoscopy (RAB) as well as in the use of advanced imaging techniques, including augmented fluoroscopy (AF) and cone beam computed tomography (CBCT) [3,4,5].

In the process of planning a pathway to the PPL using one of the computer-assisted, image-guided navigation platforms, the operator utilizes visible airways on the chest computed tomography (CT) to map a route from the PPL to the central airway. This is often known as the “bronchus sign”, which is defined as the presence of an airway leading directly to a pulmonary lesion. Several studies and meta-analysis have shown that the presence of a “bronchus sign” predicts higher diagnostic yield as compared to its absence [4,6,7,8,9]. In fact, the British Thoracic Society guidelines recommend bronchoscopic evaluation of pulmonary nodules when a “bronchus sign” is present on chest CT (Grade D Recommendation) [10]. However, the “bronchus sign” may not always be present during pre-procedural planning, especially in patients with emphysema in which the resolution of the chest CT is suboptimal for identifying peripheral airway walls. Recognizing that the branches of the pulmonary artery, lymphatics, and airways are often present together as the bronchovascular bundle in pulmonary lobular anatomy, we postulate that a branch of the pulmonary artery leading towards the pulmonary lesion (“artery sign”) could be used similarly when a “bronchus sign” is absent. In fact, Shinagawa et al. showed that the presence of either an airway or pulmonary artery branches leading directly to a target lesion on chest CT predicted a higher rate of success for CT-guided transbronchial biopsy using an ultrathin bronchoscope with virtual bronchoscopy [11]. We are aware that many operators use the vessels (segmental and subsegmental pulmonary artery branches) for planning a pathway to the desired target. However, there are no studies evaluating the feasibility and safety of using the “artery sign” for planning and performing navigation bronchoscopy.

In our clinical practice with ENB and RAB, we have been utilizing a branch of the pulmonary artery leading to the PPL as a backup pathway when the registration CT scan lacks a “bronchus sign”. In this study, we aim to describe the feasibility and safety of using the “artery sign” as a surrogate of “bronchus sign” to successfully create a path to a target lesion for PPL sampling.

## 2. Methods

### 2.1. Patients

We retrospectively reviewed the data from cases in which the “artery sign” was used for pre-procedural planning for conventional ENB (with the superDimension™ system, Medtronic, Minneapolis, MN, USA) and RAB (with Monarch™ system, Johnson & Johnson, Redwood City, CA, USA) which were performed for PPL sampling. Consecutive cases from July 2020 until July 2021 at the University of Minnesota Medical Center, Minneapolis, MN and from June 2018 until December 2019 at the University of Chicago Medical Center, Chicago, IL were included in this study. The medical records of these patients who required guided bronchoscopy (EMN and RAB with or without r-EBUS) to sample pulmonary lesions were reviewed and included in the analysis. Institutional Review Board (IRB) approval was obtained (IRB #19-0011 for the University of Chicago and IRB #00013135 for the University of Minnesota).

### 2.2. Inclusion and Exclusion Criteria

#### 2.2.1. Inclusion Criteria

Consecutive adult subjects who underwent either conventional ENB or RAB for PPL sampling under general anesthesia were included in this study. Subjects were included in the final analysis only if the “artery sign” was used during pre-procedural planning phase with either platform.

#### 2.2.2. Exclusion Criteria

Subjects were excluded from the study if inspection bronchoscopy demonstrated an endobronchial lesion that was amenable to biopsy using conventional white light bronchoscopy, if the patient was diagnosed with malignancy on endobronchial ultrasound guided transbronchial needle aspiration, or if the “bronchus sign” was seen leading directly to the target lesion during the pre-procedural planning phase.

### 2.3. Endpoints

#### 2.3.1. Primary Endpoint

Navigational Success and Radial EBUS View

“Navigational success” was defined as successfully maneuvering the bronchoscope to the target lesion based on feedback from the navigation system (alignment with the target). Whether a radial EBUS imaging confirmation was obtained at the target lesion was documented. In cases where a radial EBUS confirmation was obtained, whether the view was eccentric or concentric was reported, as per our standard operating policies.

#### 2.3.2. Secondary Endpoint

Device or procedure-related complications

Device or procedure-related complications including pneumothorax (any size, even if asymptomatic), significant airway bleeding (defined by the Nashville working group consensus statement [12], and whether the use of blood transfusion, open thoracotomy, or endobronchial blockers was required), and respiratory failure within 24 h of procedure (defined as new or increased requirement of supplemental oxygen or need for post-procedure ventilatory support, invasive or non-invasive) were reported.

### 2.4. Study Design

This study was a dual-center, multi-platform, retrospective, consecutive case series.

### 2.5. Procedure

General anesthesia with an indwelling 8.0 or 8.5 endotracheal tube was used for all procedures. Tidal volumes of 6–8 mL/kg with positive end-expiratory pressure of 5–10 cm H_2_O were used for all cases. Airway inspection using a conventional white light bronchoscope was performed prior to ENB and RAB to rule out central endobronchial lesions and aspirate secretions if present. Radial EBUS probe (Olympus UM-S20-17S; 20 MHz) was used as a confirmatory tool to verify the target lesion in all cases. Biopsy tools consisted of a transbronchial aspiration needle (Olympus Periview Flex 21G needle, Medtronic Arc point 21G needle, Bronchus 19G needle or the Auris 21G needle (prior to its withdrawal from the market)) and transbronchial biopsy forceps (Monarch™ Auris Smooth cupped forceps or superDimension™ forceps) that were advanced through the working channel to the target lesion. Rapid onsite cytology evaluation (ROSE) was performed in all cases. Diff-Quik smears were prepared in the bronchoscopy suite for needle aspirates and touch preps for transbronchial forceps biopsies.

### 2.6. Data Collection

Baseline features of study patients including age and gender were recorded. Lesion characteristics including lesion location (i.e., right upper lobe (RUL), right middle lobe (RML), right lower lobe (RLL), left upper lobe (LUL), and left lower lobe (LLL)), distance from pleura, lesion appearance (solid, ground glass, or mixed), and size were recorded. Navigation success based on feedback from the navigation system, r-EBUS image acquisition, vessel width, and procedure related adverse events were recorded.

## 3. Results

During the study period, 45 subjects underwent PPL sampling with the superDimension™ ENB platform at the University of Minnesota Medical Center and 124 subjects underwent PPL sampling with the Monarch™ RAB platform at the University of Chicago Medical Center. Of these, 30 (17.8%) patients (15 with the superDimension™ ENB platform at the University of Minnesota Medical Center, 15 with the Monarch™ RAB platform at the University of Chicago Medical Center) used the “artery sign” for pre-procedural planning in the absence of the “bronchus sign” (Figure 1).

### 3.1. Baseline Features of Study Patients

A total of 30 patients met criteria for this study. The baseline features of the included patients are presented in Table 1.

### 3.2. Lesion Characteristics

The median size of the target lesions based on the largest measurable diameter was 17 mm (6–32 mm, mean: 18.5 mm). The median distance of the target lesion to the pleura was 5 mm (0–34 mm, mean: 10.6 mm). Eleven lesions (36.7%) were solid, 15 were pure ground glass (50.0%), and 4 were mixed (13.3%) on chest CT. Eleven lesions were in the RUL (36.7%), 5 in the RML (16.7%), 3 in the RLL (10.0%), 8 in the LUL (26.7%), and 3 in the LLL (10.0%). Other lesion characteristics are listed in Table 1.

### 3.3. Pre-Procedural Planning

All cases were planned successfully using the “artery sign” on either the superDimension™ ENB platform (n = 15) or the Monarch™ RAB platform (n = 15). Successful planning was defined as the ability to connect the target lesion with a central airway by following a branch of the pulmonary artery, and not an airway. The median size of the largest cross-sectional diameter of the vessel width used for planning was 3 mm (1.0–6.6 mm, mean: 3.1 mm).

### 3.4. Procedure Data

Navigation to the target was successful for 29 of 30 lesions (96.7%) based on feedback from the navigation system (alignment with the virtual target). A r-EBUS image was acquired in 27 cases (90%) [eccentric view in 13 (43.33%) and concentric view in 14 patients (46.66%)], while in 3 cases (10%) no r-EBUS view was obtained (Table 2).

### 3.5. Adverse Events

Pneumothorax occurred in 1 case (3%), which did not require chest tube placement. Significant airway bleeding was reported in 1 case (3%) (Grade 2, as defined by the Nashville working group consensus statement [12]), which stopped after suctioning of blood with wedging of the bronchoscope into the airway segment for >1 min. There was no need for blood transfusion, open thoracotomy or use of endobronchial blockers in any cases. There were no reports of respiratory failure after the procedure. None of the cases resulted in a complication which required hospitalization (Table 3).

## 4. Discussion

Advanced navigation technologies and imaging (i.e., ENB, RAB, CBCT) seem to improve diagnostic yield for PPL sampling when compared with conventional bronchoscopy, although no direct comparative trials have been performed to date for these emerging techniques [3,4,5]. RAB platforms have been developed with the aim of increasing diagnostic yield by providing improved navigation, farther reach, and stability during lesion sampling as compared with guided bronchoscopy platforms [4,6,7,13,14].

In this consecutive case series, we used the superDimension™ system for PPL sampling in the conventional ENB cohort. The superDimension™ system is composed of four components: a location guide (LG), laptop computer with proprietary software for pre-procedural planning, an electromagnetic mat placed under the patient, and an imaging tower. For the PPL that were sampled via RAB, the Monarch™ platform by Auris Health was used. The Monarch™ platform is composed of four components: the robotic bronchoscope cart, laptop computer with proprietary software for pre-procedural planning, an electromagnetic sensor placed above the patient, and an imaging tower.

For both the ENB and RAB systems, a thin-slice protocol chest CT is required for the pre-procedural planning phase. At our institutions, we obtain a chest CT with 1.0 mm thick sections with the patient at the end of an inspiratory hold maneuver so that lung volumes are close to total lung capacity (TLC). The images are imported into the planning platform and the proprietary software converts the patient’s radiographic anatomy to virtual anatomy which allows for navigation during bronchoscopy. During the planning phase, the operator selectively chooses the airway that approaches or directly leads to the target lesion (the “bronchus sign”).

Regardless of navigation approach, in many studies, the presence of a “bronchus sign” has been associated with an improvement in diagnostic yield. In a meta-analysis of 2199 lesions, the diagnostic yield was 74.1% with the bronchus sign vs. 49.6% in its absence. Additionally, the odds ratio for successfully diagnosing a lesion with a “bronchus sign” on CT was 3.4 [6]. Specifically, for ENB, Seijo et al. reported an increased diagnostic yield of PPL sampling when there is a presence of a “bronchus sign” as opposed to its absence (79% vs. 31%) [8]. Likewise, for RAB, the diagnostic yield of PPL sampling was increased with the presence of a “bronchus sign” in a multi-center study by Chaddha et al. (78.3% vs. 54.1%) and in the BENEFIT trial (75.0% vs. 72.7%) [4,7]. The large multi-center NAVIGATE trial evaluating an EMN system by Medtronic also demonstrated that the presence of a “bronchus sign” was associated with higher diagnostic yield (78.3% vs. 67.1%) [9]. However, studies have reported that up to 40% of patients lack a “bronchus sign” when undergoing navigational bronchoscopy especially in patients with emphysema in which the resolution of the chest CT is suboptimal for identifying peripheral airway walls [8,15,16,17]. Therefore, alternative strategies are necessary to provide navigation precision.

In their study, Shinagawa et al. showed that the presence of either an airway or branch of the pulmonary artery leading directly to a target lesion on chest CT predicted a higher rate of success for CT-guided transbronchial biopsy using an ultrathin bronchoscope with virtual bronchoscopy; with diagnostic sensitivities up to 80% as compared to 14% when both features are absent [11]. Likewise, by appreciating that branches of the pulmonary artery, lymphatics and airways are adjacent in the bronchovascular bundle, we hypothesized that in patients where a branch of the pulmonary artery is seen leading to the target nodule there should also be a contiguous airway leading to the nodule. Therefore, the vessel can be used for pathway mapping to the desired target in navigational bronchoscopy.

In this retrospective series of pulmonary nodule biopsies with ENB and RAB performed at two institutions, we describe 30 cases where the target lacked a “bronchus sign” but we planned the pathway using a vessel (Figure 2). In all these cases, there was absent “bronchus sign” leading directly to the lesion, therefore the airway could not be used alone during the pre-procedural planning phase to map a pathway to the target lesion. Successful navigation to the target lesion was demonstrated in 29 of 30 (96.7%) cases based on feedback from the navigation system using the “artery sign” during pre-procedural planning. Due to the pervasive problem of CT-to-body divergence, we also evaluated the successful navigation based on radial EBUS view. This helps with proper placement of the sampling tools relative to the lesion (Figure 3). In our study, concentric views, eccentric views, and absent views on radial EBUS was obtained in 47%, 43%, and 10%, respectively.

We are aware that many bronchoscopists use the “artery sign” for planning purpose, but this has never been formally evaluated in a study. We believe that planning by using the “artery sign” may improve the rate of navigation success for ENB and RAB procedures, especially in cases without a “bronchus sign”. The theoretical evidence supporting vessels as surrogates for airways is well corroborated by developmental biology. During early fetal development, the airways act as a template for pulmonary vessel development whereby vessels form by vasculogenesis around the branching airways [18]. The human lung undergoes four stages of prenatal lung development (embryonic, pseudoglandular, canalicular and alveolar stages) and a close relationship between blood vessels and the airways are found throughout this development [19]. During the embryonic stage, the lung bud develops to form lobar and then segmental airways which are accompanied by blood supply on the ventral side of each lung bud. In the pseudoglandular stage, the lung buds further divide into pre-acinar airways and at the same time, all pre-acinar pulmonary arteries and veins are formed. More division occurs in the canalicular stage to form the respiratory airways, blood-gas barrier, and epithelial differentiation into type I and II pneumocytes. In the last stage, a double capillary wall forms and the appearance of true alveoli are present. Blood vessels develop at the same time as airways and more specifically, the pulmonary arteries run alongside the airways and the pulmonary veins show a similar branching pattern to the arteries, though separated from them. Thus, pulmonary blood vessels may be used as a surrogate for an airway when a “bronchus sign” is absent.

Adverse events (AE) related to navigational bronchoscopy typically include pneumothorax and airway bleeding. In this case series, pneumothorax occurred in 1 out of 30 patients (3.3%), which resolved spontaneously without requiring chest tube placement or hospitalization. The rate of pneumothorax in our study was comparable with what is reported in the literature [4,7,9]. Although a small sample size, it is interesting to note that the one case of pneumothorax in our case series was in the ENB cohort. We speculate that this may be related to stability of the RAB platform and the ability to wedge its sheath in a segmental or even sub-segmental airway. This may prevent any airflow and positive pressure towards the target during ventilation at the time of biopsy and reduce the rate of pneumothorax even if minimal pleural injury occurs during sampling.

In patients with vessels leading to the lesions, there is a concern for bleeding post bronchoscopic biopsy. Significant airway bleeding related to the procedure was found in 1 out of 30 patients (3.3%) in our series. The bleeding stopped after suctioning of blood with wedging of the bronchoscope into the airway segment for >1 min. There was no need for blood transfusion, open thoracotomy or use of endobronchial blockers. This airway bleeding rate is comparable with the results reported in the literature [4,7]. The low rate of airway bleeding from PPL sampling during ENB and RAB may be related to the relatively low-pressure vascular system in the distal lung. Despite a small sample size, it is also interesting to note that the one case of significant airway bleeding in our case series was also in the ENB cohort. This may again be due to the stability offered by the RAB platform and the ability to keep the Monarch™ robotic sheath (outer diameter 6.0 mm) wedged in the most distal segmental or sub-segmental airway possible. This way, if bleeding were to occur during sampling, blood clot will form around the scope or will be drained through the scope into the suction tubing instead of spilling of blood into the other segments of normal lung causing hypoxemia.

Vessel mapping has been used in a prospective, single-arm, multicentered study but for a different purpose. Sun et al. demonstrated the efficacy and safety profile of using fused fluoroscopy and vessel mapping to aid with virtual bronchoscopic navigation and pulmonary lesion sampling. The avoidance of virtually mapped blood vessels during transparenchymal nodule sampling may have contributed to their low rate of airway bleeding [20]. This may be clinically relevant, especially when sampling pulmonary lesions that are central and adjacent to larger pulmonary vessels. However, whether this can be generalized to other bronchoscopic navigation platforms remains to be studied.

### Limitations

Our study has a small sample size and evaluates the utility of the “artery sign” for only two of the available advanced bronchoscopy platforms. Indeed, the software available for the generation of virtual airways and navigation maps varies between navigation platforms. Improved segmentation with peripheral lung navigation software may not require the use of the “artery sign” if the airway is identified by the software in the proximity of most lesions. The ATLAS study, which compared airway segmentation and pathway generation in 41 PPL from 25 patients via three different planning platforms, showed significant differences among the studied software programs in regard to the distance between the terminal end of the virtual navigation pathway and the target pulmonary lesion [21]. With that said, the data in this study are applicable to selected patients undergoing nodule biopsy using ENB (superDimension™ system) and RAB (Monarch™ platform). Because of the retrospective nature of the study, the distance between the robotic scope/ENB catheter and the target is not available. From a practical standpoint, in all procedures, the scope/ENB catheter is advanced as close as possible. In fact, r-EBUS confirmation was obtained in 90% of cases, confirming that the vessel planning can lead to navigation success similar or better than in other robotic-assisted bronchoscopy studies [4,7]. Lastly, to truly show the benefit of using the “artery sign” for pre-procedural planning in navigational bronchoscopy, a prospective study will have to be designed to evaluate the yield and safety of the procedure in patients with absent “bronchus sign” but present “artery sign” as compared with patients without both features. Despite these limitations, we believe that our findings offer an alternative method to plan pathways during pre-procedural planning for navigational bronchoscopy. In this regard, the “artery sign” is a feasible option during ENB and RAB when the “bronchus sign” is absent.

## 5. Conclusions

We describe the concept of using the “artery sign” as a surrogate for “bronchus sign” when planning navigation pathways for ENB and RAB. Navigational bronchoscopy software continues to improve [21] and advanced imaging techniques continue to develop and be implemented in bronchoscopy, including augmented fluoroscopic navigation technology [22], O-arm [23] and cone-beam CT [5]. Until then, the ability of the operator to recognize and plan a pathway from the central airway to the target lesion is relevant for optimizing yield. Prospective, larger studies are needed to clarify the impact on diagnostic yield and complication rates.

## Figures and Tables

**Figure 1 diagnostics-12-03059-f001:**
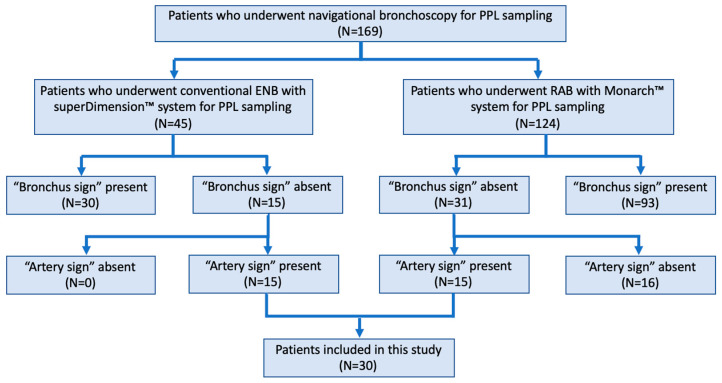
Flow diagram depicting how patients were selected for study inclusion. PPL = parenchymal pulmonary lesion; ENB = electromagnetic navigation bronchoscopy; RAB = robotic-assisted bronchoscopy. In patients with “bronchus sign” present, the presence of the “artery sign” was not evaluated since the planning software and algorithm are based on airway segmentation, and therefore the presence of the “artery sign” does not impact the workflow of the procedure.

**Figure 2 diagnostics-12-03059-f002:**
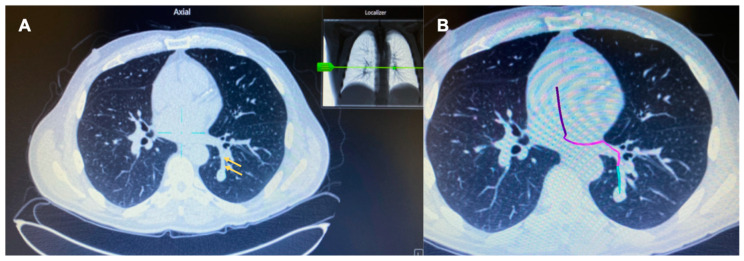
CT image in the axial plane showing the absence of an airway, but presence of a vessel (yellow arrows) leading to the target lesion in the left lower lobe (**A**). Successful mapping of a pathway to the target lesion using the “artery sign” on the on the Medtronic superDimension™ electromagnetic navigation bronchoscopy platform during pre-procedural planning (**B**).

**Figure 3 diagnostics-12-03059-f003:**
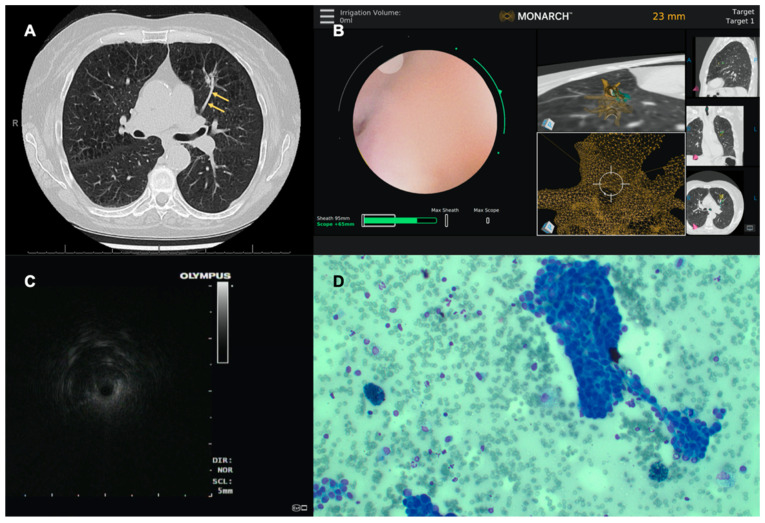
CT image in the axial plane showing the absence of an airway, but presence of a vessel (yellow arrows) leading to the target lesion in the left upper lobe (**A**). Successful navigation to the target lesion using the Monarch™ robotic platform as indicated by the alignment of the robotic scope in relation to the target lesion on the virtual target screen (**B**). An eccentric view of the target lesion is seen on radial EBUS (**C**). Needle biopsy of the lesion is performed and Diff-Quik smear from the needle aspirate shows malignant cells (**D**).

**Table 1 diagnostics-12-03059-t001:** Patient Demographics and Lesion Characteristics.

Demographics and Lesion Characteristics	N = 30
Mean Age (IQR)	68 (40–89)
Gender	
Male (%)	11 (37%)
Female (%)	19 (63%)
Lesion Location	
Right Upper Lobe (%)	11 (37%)
Right Middle Lobe (%)	5 (17%)
Right Lower Lobe (%)	3 (10%)
Left Upper Lobe (%)	8 (27%)
Left Lower Lobe (%)	3 (10%)
Distance from Pleura (mm)	5 (0–34)
Lesion appearance	
Solid (%)	11 (37%)
Ground Glass (%)	15 (50%)
Mixed (%)	4 (13%)
Lesion Size	
≤20 mm (%)	17 (57%)
21–30 mm (%)	12 (40%)
>30 mm (%)	1 (3%)

Data are presented as the mean (interquartile range), n (%), or as indicated otherwise.

**Table 2 diagnostics-12-03059-t002:** Navigation Success and r-EBUS View.

Navigation Success and r-EBUS View	N = 30
Successful Navigation (%)	29 (97%)
r-EBUS view	
Concentric (%)	14 (47%)
Eccentric (%)	13 (43%)
No View (%)	3 (10%)

Data are presented as n (%), or as indicated otherwise.

**Table 3 diagnostics-12-03059-t003:** Procedural Complications.

Complications	N = 30
Pneumothorax (%)	1 (3%)
Bleeding (%)	1 (3%) Grade 1–0 (0%) Grade 2–1 (3%) Grade 3–0 (0%) Grade 4–0 (0%)
Respiratory Failure	0 (0%)

Data are presented as n (%), or as indicated otherwise. Grade 1 Bleeding: Requiring less than 1 min of suctioning or wedging of the bronchoscope resulting in spontaneous cessation of bleeding. Grade 2 Bleeding: Suctioning more than 1 min or need for re-wedging of the bronchoscope or instillation of cold saline, vasoactive substances, or thrombogenic agents. Grade 3 Bleeding: Selective intubation with endotracheal tube or balloon/bronchial blocker for less than 20 min or premature interruption of the procedure. Grade 4 Bleeding: Persistent selective intubation > 20 min or new admission to the ICU or packed RBC transfusion or need for bronchial artery embolization or resuscitation. Standardized Definitions of Bleeding After Transbronchial Lung Biopsy: A Delphi Consensus Statement from the Nashville Working Group [12].

## Data Availability

The authors are accountable for all aspects of the work (if applied, including full data access, integrity of the data and the accuracy of the data analysis) in ensuring that questions related to the accuracy or integrity of any part of the work are appropriately investigated and resolved.

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
