# Peer review of "The Feasibility of Using the “Artery Sign” for Pre-Procedural Planning in Navigational Bronchoscopy for Parenchymal Pulmonary Lesion Sampling"

_diagnostics, 2022, doi:10.3390/diagnostics12123059_

Round 1

Reviewer 1 Report

I would like to complement the authors for validating what has been used by many pulmonologists but not backed by data, i.e. the "vessel sign" through this retrospective review. They presented the data succinctly and methodically. I believe the readers will greatly appreciate the authors' contextualization of the pulmonary embryology while discussing why using a vessel as a surrogate for an airway in peripheral lung navigation makes sense. With that said, I strongly believe the manuscript needs some revision to make it even better. Below are my suggestions to the authors:

  1. Line 23 - “superDimension™” and “MONARCH™” being trademarked names should ideally be used exactly as the owners Medtronic and J&J do. The authors may want to change it accordingly everywhere in the manuscript.

  2. Line 43 - Since authors hope to legitimize the phrase “vessel sign”, it may not be a bad idea to include it in the keywords right after “bronchus sign”.

  3. Line 58 - Please change “robotic assisted” to “robotic-assisted”, and do the same in the rest of the manuscript.

  4. Line 61-62 - “using one of the navigation or robotic platforms” may be technically better said as “using one of the computer-assisted, image-guided navigation platforms”.

  5. Line 69 - “a vessel (“vessel sign”) could be used” might make more sense as “a vessel leading towards the peripheral lesion (“vessel sign”) could be used”.

  6. Line 81-87 - The word “consecutive” has been used far too many times in this paragraph without adding additional meaning so it is best replaced/deleted in a few sentences. 

  7. Line 81-83 - Revise the starting sentence to “We retrospectively reviewed the data from the cases in which the “vessel sign” was used for pre-procedural planning for conventional ENB (with the Super Dimension system) and RAB (with Monarch system) performed for PPL sampling”.

  8. Line 85 - “consecutive cases” here is a repetition in the same sentence, and deleting it leaves the meaning unchanged, so revise this sentence to “Consecutive cases from July 2014 until July 2021 at the University of Minnesota Medical Center, Minneapolis, MN and from June 2018 until December 2019 at the University of Chicago Medical Center, Chicago, IL were included in this study”.

  9. Line 86-88 - Replace “consecutive patients” with “these patients”, “require” with “required”, and delete the comma between “pulmonary lesions, were reviewed”.

  10. Line 89-90 - The authors need to be consistent in their use of “The” in front of the names of the study sites. For example, “(IRB #19-0011 for The University of Chicago and IRB #00013135 for University of Minnesota)”. Perhaps, simplifying it as “(University of Chicago IRB #19-0011; University of Minnesota IRB #00013135)” will be better. The same change will need to be made in the abstract line 26-27.

  11. Line 92-95 - The first sentence of the inclusion criteria does not fit. This is a retrospective review, so the outcomes have already occurred and saying “were acceptable candidates to undergo” is redundant and confusing. Simply stating “Consecutive adult subjects who underwent either conventional ENB or RAB for PPL biopsy under general anesthesia were included in this study” seems more appropriate.

  12. Line 95 - In continuation with the above, the next sentence ought to address the subjects included in the final analysis, and makes more sense as “Subjects were included in the final analysis only if…”.

  13. Line 107 - Do the authors intend to suggest that a tool-in-lesion confirmation was used? If not, they need to elaborate on why “successfully maneuvering the bronchoscope and instruments to the target lesion” makes more sense than just saying “successfully maneuvering the bronchoscope to the target lesion”.

  14. Line 108 - “The presence of whether a radial EBUS image confirmation was obtained at the target lesion was documented” is better stated as “Whether a radial EBUS imaging confirmation was obtained at the target lesion was documented”.

  15. Line 109 - Here too, I recommend changing “radial EBUS view” to “radial EBUS confirmation”. 

  16. Line 109-111 - This sentence states what the authors usually do in their practice, but falls short on stating that the radial EBUS confirmation was further reported in this study as concentric and eccentric views separately. Consider a simpler statement, like “In cases where a radial EBUS confirmation was obtained, whether the view was eccentric versus concentric was also additionally reported”.

  17. Line 119 - Change “are recorded” to “were reported”. 

  18. Line 121 - The authors may want to state the study design in a full sentence instead of a partial sentence, since they have not partial sentences anywhere else in the manuscript.

  19. Line 127 - I suppose the authors used “Olympus UM-S20-17S, 20 MHz” rEBUS probe. They may want to revise “Radial EBUS probe (20MHz) (Olympus) was used” to “Radial EBUS probe (Olympus UM-S20-17S; 20 MHz) was used”.

  20. Line 133 - Add a hyphen between Diff and Quik.

  21. Line 144 - “at the two centers” in this line is redundant given that the two centers are named in the sentence, so I suggest deleting it.

  22. Line 152 - Authors may delete “(mean 68 years old, 63% female)” here, and it becomes a cleaner sentence leaving the readers to review the table below that contains all the information they need.

  23. Line 154 - Table 1 may be revised to make it more visually discernible. Can the authors italicize the subheadings?

  24. Line 154 - Table 1 may be better off without the navigation platforms’ mention since it is a “patient demographics and lesion characteristics” table. Besides, authors have duly mentioned this data in the preceding paragraph. 

  25. Line 165-167 - This line is better stated as a full sentence instead of in parentheses. Also, change “proximal airways” to “central airways” and use a comma after “following a vessel” to accentuate the point being made. Consider this, for example, “Successful planning was defined as the ability to connect the target lesion with a central airway by following a vessel, not a bronchus”.

  26. Line 206 - “in our consecutive case series” is redundant since the authors have already established at the beginning of the paragraph that they are discussing “this consecutive case series”. Please delete it.

  27. Line 207 - I suggest deleting “Similarly”, and stating “The MONARCH™ platform also has four components”.

  28. Line 211 - If authors agree, I suggest they add a sentence or two about the pre-procedural CT chest protocol used at their institutions. Doing so will make this discussion more comprehensive. 

  29. Line 242 - Add a comma after “CT-to-body divergence”.

  30. Line 243 - Since rEBUS helps “with” proper placement of the sampling tools relative to the lesion but does not “confirm” it, I recommend changing “confirm” to “with” in this sentence.

  31. Line 261 - Authors may consider adding a comma after "for ENB and RAB procedures" to accentuate the point being made (that vessel sign alone may be better altogether, but especially useful in cases without a bronchus sign).

  32. Line 280-281 - Since pneumothorax occurred in just one subject, saying “none of the patients” does not make sense. Reformatting these two sentences as “In this case series, pneumothorax occurred in 1 out of 30 patients (3.3%) which resolved spontaneously without requiring chest tube placement or hospitalization” might make more sense.

  33. 291 - “The was a Grade 2 AE” is obviously a typo. Please fix this to concur with The Nashville Statement.

  34. 300 - This mother-daughter sheath-scope configuration is unique to the MONARCH™ system and not applicable to the ION system, so the authors may want to change this line as “ability to keep the MONARCH™ robotic sheath”.

  35. 321 - The authors probably meant to say “studied software programs”, not “studies software programs”. Please fix the typo.

  36. 322 - The authors probably meant to say “and the target peripheral lesion”, not “to the target peripheral lesion”. They are reporting on distance “between” point A and B, not “from point A to B”.

Author Response

  1. Line 23 - “superDimension™” and “MONARCH™” being trademarked names should ideally be used exactly as the owners Medtronic and J&J do. The authors may want to change it accordingly everywhere in the manuscript.

Reply: Thank you for pointing this out. This has been addressed on page 1,

line 24, as well as throughout the manuscript.

  1. Line 43 - Since authors hope to legitimize the phrase “vessel sign”, it may not be a bad idea to include it in the keywords right after “bronchus sign”.

Reply: Agreed. “Vessel sign” and “artery sign” have been added as keywords

on page 1, lines 44-45.

  1. Line 58 - Please change “robotic assisted” to “robotic-assisted”, and do the same in the rest of the manuscript.

Reply: Good point. The hyphen has been added, so that it now reads “robotic-

assisted”. This has been addressed on page 2, line 102, as well as throughout the manuscript

  1. Line 61-62 - “using one of the navigation or robotic platforms” may be technically better said as “using one of the computer-assisted, image-guided navigation platforms”.

Reply: Agreed. This has been changed and reflected on page 2, lines 105-106, which now reads “… using one of the computer-assisted, image-guided navigation platforms…”.

  1. Line 69 - “a vessel (“vessel sign”) could be used” might make more sense as “a vessel leading towards the peripheral lesion (“vessel sign”) could be used”.

Reply: We concur. This makes the sentence more specific. This is now

reflected on page 2, lines 117-118, which now reads “… we postulate that a

branch of the pulmonary artery leading towards the pulmonary lesion (“artery

sign”) could be used…”.

  1. Line 81-87 - The word “consecutive” has been used far too many times in this paragraph without adding additional meaning so it is best replaced/deleted in a few sentences. 

Reply: We agree. The changes are now reflected on page 2, lines 134-143.

  1. Line 81-83 - Revise the starting sentence to “We retrospectively reviewed the data from the cases in which the “vessel sign” was used for pre-procedural planning for conventional ENB (with the Super Dimension system) and RAB (with Monarch system) performed for PPL sampling”.

Reply: Agreed. This change is reflected on page 2, lines 134-136.

  1. Line 85 - “consecutive cases” here is a repetition in the same sentence, and deleting it leaves the meaning unchanged, so revise this sentence to “Consecutive cases from July 2014 until July 2021 at the University of Minnesota Medical Center, Minneapolis, MN and from June 2018 until December 2019 at the University of Chicago Medical Center, Chicago, IL were included in this study”.

Reply: We agree. Thank you for your suggestions. The changes are reflected on page 2, lines 136-139.

  1. Line 86-88 - Replace “consecutive patients” with “these patients”, “require” with “required”, and delete the comma between “pulmonary lesions, were reviewed”.

Reply: Thank you for pointing this out. The changes are reflected on page 2, lines 139-140, which now reads “… medical records of these patients who required guided bronchoscopy…”.

  1. Line 89-90 - The authors need to be consistent in their use of “The” in front of the names of the study sites. For example, “(IRB #19-0011 for The University of Chicago and IRB #00013135 for University of Minnesota)”. Perhaps, simplifying it as “(University of Chicago IRB #19-0011; University of Minnesota IRB #00013135)” will be better. The same change will need to be made in the abstract line 26-27.

Reply: This is now addressed on page 2, lines 142-143, as well as the rest of

the manuscript.

  1. Line 92-95 - The first sentence of the inclusion criteria does not fit. This is a retrospective review, so the outcomes have already occurred and saying “were acceptable candidates to undergo” is redundant and confusing. Simply stating “Consecutive adult subjects who underwent either conventional ENB or RAB for PPL biopsy under general anesthesia were included in this study” seems more appropriate.

Reply: Good point. This change is reflected on page 3, lines 200-201.

  1. Line 95 - In continuation with the above, the next sentence ought to address the subjects included in the final analysis, and makes more sense as “Subjects were included in the final analysis only if…”.

Reply: Agreed. This change is reflected on page 3, lines 201-203.

  1. Line 107 - Do the authors intend to suggest that a tool-in-lesion confirmation was used? If not, they need to elaborate on why “successfully maneuvering the bronchoscope and instruments to the target lesion” makes more sense than just saying “successfully maneuvering the bronchoscope to the target lesion”.

Reply: Thank you for pointing this out. This change is reflected on page 3, lines 213-214, which now reads “… defined as successfully maneuvering the bronchoscope to the target lesion based on feedback from the navigation system…”.

  1. Line 108 - “The presence of whether a radial EBUS image confirmation was obtained at the target lesion was documented” is better stated as “Whether a radial EBUS imaging confirmation was obtained at the target lesion was documented”.

Reply: This change is now reflected on page 3, lines 215-216.

  1. Line 109 - Here too, I recommend changing “radial EBUS view” to “radial EBUS confirmation”. 

Reply: The change is reflected on page 3, line 216.

  1. Line 109-111 - This sentence states what the authors usually do in their practice, but falls short on stating that the radial EBUS confirmation was further reported in this study as concentric and eccentric views separately. Consider a simpler statement, like “In cases where a radial EBUS confirmation was obtained, whether the view was eccentric versus concentric was also additionally reported”.

Reply: We agree. This does help the sentence flow better. This change is now

reflected on page 3, lines 216-217, which now reads “In cases where a radial

EBUS confirmation was obtained, whether the view was eccentric or

concentric was reported…”.

  1. Line 119 - Change “are recorded” to “were reported”. 

Reply: This change is reflected on page 3, line 225.

  1. Line 121 - The authors may want to state the study design in a full sentence instead of a partial sentence, since they have not partial sentences anywhere else in the manuscript.

Reply: The statement is now changed to a full sentence as reflected on page

3, line 228, which now reads “The study was a dual-center, multi-platform,

retrospective, consecutive case series.”.

  1. Line 127 - I suppose the authors used “Olympus UM-S20-17S, 20 MHz” rEBUS probe. They may want to revise “Radial EBUS probe (20MHz) (Olympus) was used” to “Radial EBUS probe (Olympus UM-S20-17S; 20 MHz) was used”.

Reply: Thank you for this clarification. The change is now reflected on page 3,

line 234.

  1. Line 133 - Add a hyphen between Diff and Quik.

Reply: A hyphen is now added (page 3, line 241), as well as throughout the

manuscript.

  1. Line 144 - “at the two centers” in this line is redundant given that the two centers are named in the sentence, so I suggest deleting it.

Reply: We agree. This change is reflected on page 4, line 290.

  1. Line 152 - Authors may delete “(mean 68 years old, 63% female)” here, and it becomes a cleaner sentence leaving the readers to review the table below that contains all the information they need.

Reply: Good point. This is change is reflected on page 4, line 304, which now

reads “A total of 30 patients met criteria for this study.”.

  1. Line 154 - Table 1 may be revised to make it more visually discernible. Can the authors italicize the subheadings?

Reply:  The table is revised with the appropriate indentation and italics to help

make the data appear more discernible (pages 4-5, lines 306-320).

  1. Line 154 - Table 1 may be better off without the navigation platforms’ mention since it is a “patient demographics and lesion characteristics” table. Besides, authors have duly mentioned this data in the preceding paragraph. 

Reply: We agree. This is now removed from the table (pages 4-5, lines 306-

320).

  1. Line 165-167 - This line is better stated as a full sentence instead of in parentheses. Also, change “proximal airways” to “central airways” and use a comma after “following a vessel” to accentuate the point being made. Consider this, for example, “Successful planning was defined as the ability to connect the target lesion with a central airway by following a vessel, not a bronchus”.

Reply: Great suggestion. Thanks. The change is now made on page 5, lines

330-332, which now reads “Successful planning was defined as the ability to

connect the target lesion with a central airway by following a branch of the

pulmonary artery, and not an airway.”.

  1. Line 206 - “in our consecutive case series” is redundant since the authors have already established at the beginning of the paragraph that they are discussing “this consecutive case series”. Please delete it.

Reply: We agree. The changes are reflected on page 7, line 404.

  1. Line 207 - I suggest deleting “Similarly”, and stating “The MONARCH™ platform also has four components”.

Reply: Thank you. This change is reflected on page 7, line 405.

  1. Line 211 - If authors agree, I suggest they add a sentence or two about the pre-procedural CT chest protocol used at their institutions. Doing so will make this discussion more comprehensive. 

Reply: Thank you for this suggestion. We have added a few sentences to illustrate our pre-procedural chest CT protocol at our institutions. This addition is reflected on page 7, lines 409-411, which now reads “At our institutions, we obtain a chest CT with 1.0 mm thick sections with the patient at the end of an inspiratory hold maneuver so that lung volumes are close to total lung capacity (TLC).”.

  1. Line 242 - Add a comma after “CT-to-body divergence”.

Reply: Thank you for pointing this out. A comma is added on page 8, line 458.

  1. Line 243 - Since rEBUS helps “with” proper placement of the sampling tools relative to the lesion but does not “confirm” it, I recommend changing “confirm” to “with” in this sentence.

Reply: Great point. This change is reflected on page 8, line 459, which now

reads “This helps with proper placement…”.

  1. Line 261 - Authors may consider adding a comma after "for ENB and RAB procedures" to accentuate the point being made (that vessel sign alone may be better altogether, but especially useful in cases without a bronchus sign).

Reply: We agree with this suggestion. This change is reflected on page 9, line

509.

  1. Line 280-281 - Since pneumothorax occurred in just one subject, saying “none of the patients” does not make sense. Reformatting these two sentences as “In this case series, pneumothorax occurred in 1 out of 30 patients (3.3%) which resolved spontaneously without requiring chest tube placement or hospitalization” might make more sense.

Reply: Thank you for catching this. We made the change on page 9, lines 528-529, which now reads “…pneumothorax occurred in 1 out of 30 patients (3.3%), which resolved spontaneously without requiring chest tube placement or hospitalization.”.

  1. 291 - “The was a Grade 2 AE” is obviously a typo. Please fix this to concur with The Nashville Statement.

Reply: Thank you for pointing this out. This sentence was removed to improve

the flow of the paragraph. The change is reflected on page 10, line 547-548, which now reads “Significant airway bleeding related to the procedure was found in 1 out of 30 patients (3.3%) in our series.”.

  1. 300 - This mother-daughter sheath-scope configuration is unique to the MONARCH™ system and not applicable to the ION system, so the authors may want to change this line as “ability to keep the MONARCH™ robotic sheath”.

Reply: Agreed that this needs to be specified. The change is reflected on page

10, line 556.

  1. 321 - The authors probably meant to say “studied software programs”, not “studies software programs”. Please fix the typo.

Reply: Thank you for catching this typo as well. This change is reflected on

page 10, line 577.

  1. 322 - The authors probably meant to say “and the target peripheral lesion”, not “to the target peripheral lesion”. They are reporting on distance “between” point A and B, not “from point A to B”.

Reply: We agree with the comment. The change is reflected on page 10, line 578, which now reads “… the terminal end of the virtual navigation pathway and the target pulmonary lesion…”.

We genuinely thank the Reviewer for their careful read of our paper and for their insightful suggestions.

Reviewer 2 Report

The authors present a multicenter retrospective analysis of 30 cases performing ENB or RAB using “vessel sign” for pre-procedural planning in the diagnosis of PPLs without bronchus sign. I think very interesting and remarkable topic. However, many problems need to be resolved, please refer to the comments below.

The topic of EBUS-TBNA would be unnecessary for Introduction. Rather, bronchus sign requires in-depth discussion in this section. Not enough description of the background and too few references cited in this section.

I have a suspicion that the term “vessel sign” may be inappropriate. Shinagawa et al. have already named a similar concept “artery sign”.

Shinagawa N, Yamazaki K, Onodera Y, et al. Factors related to diagnostic sensitivity using an ultrathin bronchoscope under CT guidance. Chest 2007;131:549–53.

Please provide the thickness of CT used for planning. The presence or absence of bronchus sign may be determined differently depending on CT imaging conditions.

In the Results, please provide a flow diagram to show how the 304 subjects were excluded. Were there no PPLs without bronchus sign in which "vessel sign" was not used for pre-procedural planning? By comparing these to 30 PPLs in this study, the utility of the "vessel sign" may be demonstrated.

Author Response

The topic of EBUS-TBNA would be unnecessary for Introduction. Rather, bronchus sign requires in-depth discussion in this section. Not enough description of the background and too few references cited in this section.

Reply: Thank you for this suggestion to make the Introduction text more concise. We have removed the sentences regarding EBUS-TBNA and elaborated on the definition and value of the “bronchus sign”, along with the appropriate references on page 2, lines 109-113, which now reads “Several studies and meta-analysis have shown that the presence of a “bronchus sign” predicts higher diagnostic yield as compared to its absence [4,6-9]. In fact, the British Thoracic Society guidelines recommend bronchoscopic evaluation of pulmonary nodules when a “bronchus sign” is present on chest CT (Grade D Recommendation) [10].”.

I have a suspicion that the term “vessel sign” may be inappropriate. Shinagawa et al. have already named a similar concept “artery sign”.

Shinagawa N, Yamazaki K, Onodera Y, et al. Factors related to diagnostic sensitivity using an ultrathin bronchoscope under CT guidance. Chest 2007;131:549–53.

Reply: We thank you for highlighting this reference. We have carefully reviewed the study by Shinagawa et al. Indeed, the “CT-artery sign” described by them is identical to the “vessel sign” that we are referring to in our manuscript. Therefore, we have changed the term “vessel sign” to “artery sign” throughout the abstract and manuscript, including the title. Obviously, our study and the Shinagawa study have different aims as we are evaluating the practical use of this sign for pathway mapping in navigation bronchoscopy. We have now modified the text to acknowledge the article by Shinagawa et al. Changes are reflected on page 2, lines 119-122, which reads, “In fact, Shinagawa et al. showed that the presence of either an airway or pulmonary artery branches leading directly to a target lesion on chest CT predicted a higher rate of success for CT-guided transbronchial biopsy using an ultrathin bronchoscope with virtual bronchoscopy [11].” and page 8, lines 441-445, which reads “In their study, Shinagawa et al. showed that the presence of either an airway or branch of the pulmonary artery leading directly to a target lesion on chest CT predicted a higher rate of success for CT-guided transbronchial biopsy using an ultrathin bronchoscope with virtual bronchoscopy; with diagnostic sensitivities up to 80% as compared to 14% when both features are absent [11].”.

Please provide the thickness of CT used for planning. The presence or absence of bronchus sign may be determined differently depending on CT imaging conditions.

Reply: Thank you for pointing this out. We agree that this detail should be included. This change is now made on page 7, lines 409-411, which reads “At our institutions, we obtain a chest CT with 1.0 mm thick sections with the patient at the end of an inspiratory hold maneuver so that lung volumes are close to total lung capacity (TLC).”.

In the Results, please provide a flow diagram to show how the 304 subjects were excluded. Were there no PPLs without bronchus sign in which "vessel sign" was not used for pre-procedural planning? By comparing these to 30 PPLs in this study, the utility of the "vessel sign" may be demonstrated.

Reply: Thank you for your comment. Your remarks made us review our dataset again. We realized that we made a mistake in our reporting of the data from the University of Minnesota. While between 2014 and 2021 there were 180 superDimension cases that occurred at the University of Minnesota, the data analyzed for “bronchus sign” and “artery sign” was only for patients who underwent superDimension procedures between 2020 and 2021 for a total of 45 cases. This error is now corrected in the Methods (page 2, line 137) and Results (page 4, line 290) sections, and reflected in the flow diagram that the Reviewer had suggested (page 4, lines 297-302). While the actual total number of patients with the “artery sign” is not changed, the denominator is different from the original submission and we made the necessary corrections.

We have now included a flow diagram and categorization based on the presence and absence of the “bronchus sign” and “artery sign” (page 4, lines 297-302). In patients with the “bronchus sign” present, the ”artery sign” was not evaluated and captured in our study since the planning software and algorithm are based on airway segmentation, and therefore the presence of the “artery sign” does not impact the workflow of the procedure. We completely agree with the reviewer that to truly show the benefit of the “artery sign”, a prospective study will have to be designed to evaluate the yield and safety of the procedure in patients with absent “bronchus sign” but present “artery sign” as compared with patients without both of these features. We do agree that this will be valuable information and should be evaluated in future studies. We acknowledge this limitation of our study in the Discussion section (page 10, lines 586-589) by inserting the following sentence:

“Lastly, to truly show the benefit of using the “artery sign” for pre-procedural planning in navigational bronchoscopy, a prospective study will have to be designed to evaluate the yield and safety of the procedure in patients with absent “bronchus sign” but present “artery sign” as compared with patients without both features.”

We genuinely thank the Reviewer for their careful read of our paper and for their insightful suggestions.

Round 2

Reviewer 2 Report

Thank you very much for your sincere response to my comments.

I have no further comments.

Congratulations and keep up the good work.

Kind regards.